# Foam Rolling Elicits Neuronal Relaxation Patterns Distinct from Manual Massage: A Randomized Controlled Trial

**DOI:** 10.3390/brainsci11060818

**Published:** 2021-06-20

**Authors:** Yann Kerautret, Aymeric Guillot, Sébastien Daligault, Franck Di Rienzo

**Affiliations:** 1Laboratoire Interuniversitaire de Biologie de la Motricité, Université Claude Bernard Lyon 1, EA 7424, F-69622 Villeurbanne, France; yann@capsix-robotics.com (Y.K.); aymeric.guillot@univ-lyon1.fr (A.G.); 2CAPSIX, 69002 Lyon, France; 3Institut Universitaire de France, F-75000 Paris, France; 4CERMEP, Imagerie du Vivant, F-69500 Bron, France; daligault@cermep.fr

**Keywords:** electroencephalography, electrodermal activity, flexibility, alertness

## Abstract

The present double-blinded, randomized controlled study sought to compare the effects of a full-body manual massage (MM) and a foam rolling (FR) intervention on subjective and objective indexes of performance and well-being. A total of 65 healthy individuals were randomly allocated to an FR, MM, or a control group who received a cognitively oriented relaxation routine. Self-report ratings of perceived anxiety, muscle relaxation, and muscle pain were used to index changes in affect and physical sensations. The sit-and-reach and toe-touch tests, as well as a mental calculation task, were used to index motor and cognitive performances, respectively. We also conducted resting-state electroencephalography and continuous skin conductance recordings before and after the experimental intervention. Both FR and MM groups exhibited neural synchronization of alpha and beta oscillations during the posttest. Skin conductance increased from the pretest to the posttest in the relaxation group, but decreased in the FR group. All interventions improved range of motion, although only the MM group outperformed the relaxation group for the toe-touch performance. MM was associated with reduced muscle pain and increased muscle relaxation. Reduced perceived anxiety after the intervention was observed in the FR group only. Overall, MM and FR both improved objective and subjective indexes of performance and well-being. Differences between the two massage interventions are discussed in relation to the effects of pressure stimulation on autonomic regulations and the proactive vs. retroactive nature of FR, compared to MM.

## 1. Introduction

Manual massage (MM) refers to a wide range of techniques such as effleurage, petrissage, tapping, friction, and vibration yielding pressures on soft tissues [1,2,3]. MM has ancestral roots across human civilizations [4]. Beck [5] underlined three main purposes to MM. First, MM can be used to promote a psychophysiological state of relaxation. Second, MM can be used to benefit a variety of health outcomes (e.g., management of pain symptoms). Eventually, MM can be used as a conditioning technique to promote motor performance and recovery [6,7].

MM requires a trained physiotherapist, and only one person can be massaged at the same time, which makes MM a difficult intervention to scale [7]. Additionally, the physiotherapist regulates subtle parameters while performing the massage routine and integrates with real-time patient feedback to adjust his/her gestures. For these reasons, it appears difficult to standardize MM interventions in experimental research protocols [7]. During MM, the massaged person has no control over massage gestures as MM puts the individual in a passive state. Conversely, self-myofascial release techniques, which consist in the self-administration of mechanical pressures on soft tissues using a tool such as a foam roller or a roller massager, require voluntary movement. There is thus a fundamental distinction between MM and self-myofascial release techniques such as foam rolling (FR). MM engages participants in a primarily retroactive mode of action control, whereas FR emphasizes proactive modes of action control (for a comprehensive review, see [7]). FR has gained popularity in sports and fitness communities and consists of back and forth movements, eliciting pressures and frictions on the soft tissues [7,8,9,10,11]. As MM, FR stimulates proprioceptive organs such as the Golgi tendon organ, Pacinian, Meissner, and Ruffini corpuscles, and interstitial type III and IV receptors [12,13,14]. Both FR and MM elicit muscle tone relaxation patterns similar to those elicited by myofascial manipulations [14,15]. It is thus suggested that these techniques have the potential to impact autonomic regulations. Nonetheless, presumably due to methodological limitations, direct investigations of the mechanisms underlying the effects of FR remain sparse [7].

Behavioral data suggest that both FR and MM have beneficial effects on range of motion and contribute to alleviate muscle pain [7,9,10,11,16,17]. Both techniques were found to be associated with decreased tissue stiffness, fibrous adhesions, and scar tissue [18]. Contrary to passive stretching, MM and FR practice during warm-up and recovery routines did not hamper subsequent physical performances [6,7]. They contributed to reduce anxiety, depression, and attenuate hormonal markers of stress responses, as indicated by the concomitant decrease in cortisol and increase in dopamine/serotonin concentrations after MM [19,20,21]. Heart rate variability measures confirmed a shift from sympathetic to parasympathetic dominance in autonomous nervous system activity as a result of MM and FR [22,23,24,25]. The effects of MM eventually appeared to improve cognitive performances, possibly due to a more relaxed and alert psychophysiological state. After a 5-week intervention of MM, Field et al. [19] measured an anxiety reduction and an improvement in mental calculation performances. These results suggest that massage interventions have the potential to affect cognitive states. Using electroencephalography (EEG), Jodo [26] investigated, in a pioneering study, the effects of a facial massage on electric brain potentials. MM was found to be associated with decreased alpha and beta power (see also [19]). MM might thus have elicited a pattern of enhanced alertness, while a pattern of drowsiness was observed in the autogenic training and control groups. Comparable brain rhythms and psychological factors were elicited in elderly people after hand and foot massage [27]. However, conflicting results were found depending on the pressure level and massage type. Diego et al. [28] emphasized that a light pressure massage increased arousal response, while a moderate pressure massage was associated with lower arousal and relaxation. Surprisingly, a comparable relaxing effect also occurred after a unilateral massage [29]. However, to the best of our knowledge, whether such effects might also be obtained after FR remains unknown. This issue is of specific interest considering the proactive nature of FR, compared to MM, with regard to motor control strategies primed for the participant throughout the massage interventions.

In the present experiment, we compared the effects of MM and FR on objective and subjective indexes of performance and well-being. We implemented resting-state EEG measures to investigate the effects of these two interventions on electric brain potentials. We hypothesized that both MM and FR would impact autonomic nervous system regulations through bottom-up regulatory processes in response to the stimulation of proprioceptive organs. Comparable relaxation brain states were expected in the resting state as a result of FR and MM since both techniques are likely to involve comparable pressures on soft tissues. Yet, we hypothesized that improvements in cognitive performances would be reduced after FR. Indeed, the physical and cognitive demands associated with the proactive nature of FR might interfere with the optimal psychophysiological relaxation state.

## 2. Materials and Methods

### 2.1. Participants

In total, 65 healthy adults volunteered to participate in the present experiment (Table 1). Recruitment strategies consisted of poster advertising and personal contact between CrossFit^®^ coaches and practitioners. Inclusion criteria were the regular practice of FR (10 min routines, 2–3 times per week over the past 6 months) to ensure proper execution of techniques, especially the pressure level. Participants were free from any chronic medical conditions (including functional limitations) that might affect proprioception and/or pain perception [30]. They were also asked to refrain from physical activity, alcohol, and caffeinated beverages 24 h prior to testing. All participants signed a written informed consent form prior to enrolment. The experiment was approved by the local ethics committee (IRB 2019-A01732-55), and conducted in accordance with the ethical standards laid down in the Declaration of Helsinki and its later amendments [31].

### 2.2. Experimental Design

The present double-blinded, randomized controlled trial involved two experimental sessions (Figure 1A) supervised by the same experimenter over an inclusion period of 6 months. Participants first completed a familiarization session before being randomly allocated to one of the experimental groups: (i) relaxation (*n* = 25), (ii) foam rolling (*n* = 20), and (iii) manual massage (*n* = 20) (see below for further descriptions). Then, pretest and posttest measures were implemented. Testing was conducted between 12 a.m. and 6 p.m. The assessor was blinded to the group assignment and remained the same throughout the study to ensure reproducibility of assessment methods.

#### 2.2.1. Familiarization

The familiarization session took place 2 days before the experimental session and was used to collect demographic information (Table 1). The experimenter demonstrated the range of motion tests and neurophysiological measurement procedures, provided instructions regarding the subjective ratings, and explained the mental calculation task. The purpose of this familiarization was to minimize potential errors and save time between measurements on the day of the experimental session.

#### 2.2.2. Foam Rolling Intervention

Participants in the FR group completed a 16 min routine using two commercial tools: a foam roller of 15 × 30 cm length (Blackroll, Bottighofen, Switzerland), and a lacrosse ball (WODFitters, USA). Participants were instructed to perform back and forth movements at a rate of one inch per second, for 60 s and for each target muscle (i.e., feet plantar muscles, triceps surae, hamstrings, quadriceps, hip adductors, iliotibial band, gluteus, inferior and superior spine region). The timing was externally cued by an auditory device to ensure reproducibility. In keeping with the methods emphasized in previous FR experiments [7,10], participants were instructed to regulate the rolling pressure so that the perceived pain would not exceed 7 on a numerical rating scale (NRS) ranging from 0 (“Absence of pain”) to 10 (“Worst possible pain”). The FR routine consisted of unilateral exercises alternatively performed on both legs and bilateral exercises on lower and back muscles. All participants started with their feet in a stand-up position with a lacrosse ball. Then, participants performed on the ground the FR exercises with the foam roller targeting leg and back muscles. FR movements were first performed in a seated position between the Achilles tendon and the popliteal fossa for the triceps surae. Then, between the popliteal fossa and the ischial tuberosity to target hamstrings muscles. Then, participants rolled the quadriceps, hip adductors, and iliotibial band in plank position before returning to a seated position to apply FR to gluteus muscles, between the posterior superior iliac spine and the gluteal fold. They then had to continue the session by lying on their back, arms crossed over their chest, to target superior spine muscles (T9–T1). The last part of the routine consisted of FR inferior spine muscles (T10–L5). To limit excess pressure on the lower back, the exercise was performed in a stand-up position with the foam roller placed between the participant’s back and a wall [33]. The choice of these treatment instructions was based on the systematic review of past experiments in the field to disclose invariants in FR methods [7,34,35,36,37].

#### 2.2.3. Manual Massage Intervention

Participants in the manual massage group received MM by a professional physiotherapist who was blinded to the purpose of the experiment (20 years of registered practice). Participants were lying in the prone position on a massage table. To facilitate the physiotherapist’s maneuvers, massage oil was used. The MM protocol reproduced several features of the FR intervention, i.e., order of the areas massaged, massage time per muscle, and total duration of the session. Obviously, it was not possible to control pressures applied by the physiotherapist, who was explicitly encouraged to apply usual pressure from classical practice to improve the ecological validity of the findings. The physiotherapist declared that massage pressures were adjusted according to the zones of perceived tension and remained consistent across participants, to fit with real-life massage scenarios.

#### 2.2.4. Relaxation Intervention

Participants in the relaxation group listened to a 16 min relaxation audio soundtrack, based on the autogenic training method [38]. Participants were lying on the massage table in a supine position, wearing headphones (Beats Electronics, Culver City, CA, USA). During this desensitization–relaxation technique, they were guided to visualize bodily perceptions of heaviness and warmth in their arms, thorax, and legs. To facilitate concentration, participants were left alone in a dark room.

### 2.3. Dependent Variables

We focused on six dependent variables, all of which were collected before (pretest) and after (posttest) the experimental interventions (Figure 1B). Changes in perceived anxiety and physical sensations (muscle relaxation, warmth, and pain) were collected using Likert-type scales. Physical performances were measured using the sit-and-reach and toe-touch tests. A mental calculation task was used to index cognitive performances. Brain activity was investigated using resting-state EEG and autonomic nervous system activation using skin conductance (SC).

#### 2.3.1. Self-Report Ratings on Likert-Type Scales

Participants were requested to provide self-report ratings on a Likert-type scale ranging from 0 (“Not at all”) to 10 (“Very strong”). They provided subjective ratings of their perceived general state of anxiety (“How anxious are you feeling right now?”), and their perceived muscle sensations. We first indexed the subjective perception of muscle relaxation (“What is your level of overall muscle relaxation right now?”), warmth (“What is your level of overall muscle warmth?”), and pain (“What is your level of overall muscle pain?”).

#### 2.3.2. Motor and Cognitive Performances

We evaluated the range of motion of lower back and hamstrings muscles using the sit-and-reach test (SRT), followed by the toes touch (TT) test [39,40]. For the SRT, participants adopted a seated position with their feet, hip-width apart, and the knees extended. Then, they were instructed to lean forward slowly and gradually, without bouncing, and try to reach as far as possible with their hands on the tape measure. For the TT test, participants started from the stance on a wooden box. Legs were outstretched with feet hip-width apart. Unlike the SRT, they had to tilt the chest forward using gravity. The best score of the three trials was recorded to the nearest 0.1 cm, for both ranges of motion tests. For the trial attempt, they were instructed to hold their maximal stretching position for 2 s [41].

To index cognitive performances, we used a mental calculation task. Performance in mental calculation paradigms was positively affected in past research by massage therapy, aromatherapy, or Tai chi/yoga interventions [19,42,43]. The task consisted of iterated subtractions (−7) from a starting number randomly selected between 500 and 1000. The participants had 120 s to execute the maximal number of correct subtractions (i.e., the total number of correct answers). To avoid contamination effects on other dependent variables, including EEG and electrodermal recording, this stressful task was performed at the end of the pretest and posttest assessments [44].

#### 2.3.3. Electroencephalography

Electric brain potentials were measured at rest using EEG (rsEEG). Brain electric potentials were recorded for 4 min, during both the pretest and the posttest. To limit artifacts caused by postural changes and eye movements, participants seated in a chair and kept their eyes open, staring at a cross mark placed at sight’s height. They were asked to focus on the cross mark while remaining motionless, limiting eye blinks, and not focusing their attention on anything specific for 4 min. EEG was recorded using 16 Ag-AgCl electrodes attached on a Lycra stretchable cap according to the international 10–20 system (MLAEC2 Electro-cap system 2, ADInstrument, Dunedin, New Zealand). Electrode gel was injected to keep the impedance below 5 kΩ at the following scalp sites: FP1, FP2, F3, Fz, F4, C3, Cz, C4, TP7, CP3, CP4, TP8, P3, and P4 sensors. Two additional electrodes were positioned as reference points to facilitate artifact scoring, one on the right internal canthus and the second on the right acromion. All electrodes were connected to the input box of the EEG recording system, two Octal Bio Amp (FE238, Octal Bio Amp, ADInstrument, Dunedin, New Zealand) and one Power Lab 16/35 data acquisition system (PL3516, Power Lab 16/35, ADInstrument, Dunedin, New Zealand). The EEG software used in this experiment was LabChart Pro (ADInstrument, Dunedin, New Zealand).

Data preprocessing was performed using Brainstorm [45], which is documented and freely available for download online under the GNU general public license (https://neuroimage.usc.edu/brainstorm, accessed on 18 June 2021). EEG data were first referenced relative to the Cz electrode and a bandpass filter was applied (0.5–60 Hz). Brainstorm algorithms were then applied to automatically detect signal portions containing muscle and eye blinks activities. Visual inspection of the results of the automatic detection phase completed the artifact detection procedure. rsEEG recordings were finally epoched in 1 s time windows. Epochs containing artifacts were removed from the following steps of data analysis (i.e., 11.2% rejection rate).

We calculated the power spectrum densities for each epoch in the alpha (8–12 Hz) and beta (13–30 Hz) frequency bands (Welch’s method, fixed time window). These frequency domains were emphasized for their relevance in the study of sensorimotor processes [46,47]. Power spectrum densities were finally averaged for each participant and experimental condition to quantify rsEEG power over the course of the 4 min. We were specifically interested in power changes from the pretest to the posttest within the predetermined frequency domains of interest across the resting-state time window. Neural synchronizations corresponds to an increase in signal power for a given frequency range, whereas desynchronization patterns (i.e., desynchronization) or inhibition (synchronization) decrease [48,49].

#### 2.3.4. Electrodermal Activity

We investigated the electrodermal activity from continuous SC recordings. Two-finger electrodes (MLT116F, GSR Finger Electrodes, ADInstruments, Dunedin, New Zeland) with constant voltage (0.5 V) were positioned on the second phalanx of the second and third fingers of the right hand. Signals were processed online using a galvanic skin resistance amplifier (FE116 GSR Amp, ADInstruments, Dunedin, New Zeland) connected to a PowerLab 16/35 acquisition system. Data were coregistered with EEG signals using the LabChart Pro acquisition software. SC mirrors the activity of eccrine sweat glands, which are under the unique control of the sympathetic branch of the autonomous nervous system. Tonic changes reflect variations in physiological arousal, whereas phasic changes primarily reflect cognitive changes such as increased vigilance during motor preparation [50]. Here, we were interested in participants’ physiological arousal during the pretest and posttest resting-state measures. Hence, we first recorded the baseline SC, i.e., before the onset of the resting state period. Then, we continuously recorded SC over the 4 min resting-state period. To index the general physiological arousal, we finally calculated a normalized SC index (SCNorm) according to the following formula:(1)$$SC_{Norm}=\frac{mean\left(Resting\;State\;SC\right)}{Baseline\;SC}$$

## 3. Statistical Analysis

### 3.1. Power/Sample Size Considerations

We did not run a priori power calculation. We determined the sample size based on previous trials addressing the efficacy of self-myofascial release interventions [26,28,29]. We accounted for the fact that randomized control trial designs involving between-group comparisons usually require larger sample sizes (4–8 fold) than designs involving within-group comparisons to achieve the same statistical power [51]. Randomized control trial designs investigating the effects of FR usually involve 10–15 participants [52,53,54]. Hence, 65 participants (*n* > 20 for each experimental group) was expected to constitute a reliable sample size. To overcome remaining power limitations, we ran a posteriori power calculations for statistically significant main and interaction effects using the pwrpackage in R [55].

### 3.2. Randomization

We performed group assignments using full-blind randomization, i.e., simple randomization [56]. We generated random allocation sequences in males and female participants. To grant allocation concealment, the random sequence was matched to the predetermined list of male and female participants corresponding to their inclusion order.

### 3.3. Data Analysis

We used nlme [57] to run a linear mixed-effects analysis (with by-subjects random intercepts) of the dependent variables. We built the random-coefficient regression models and entered GROUP (foam rolling, manual massage, relaxation) and TEST (pretest, posttest) as the fixed effects (with interaction term). For the range of motion data, we included the fixed effects of TEST TYPE (toe touch, sit-and-reach) in the model. For changes in affect and physical pain measures, we, respectively, added the fixed effect of VARIABLE TYPE (i.e., muscle pain, muscle relaxation, muscle warmth). For the EEG power analysis, we included the fixed effects of SENSOR (F3, F4, C3, C4, P3, P4) and FREQUENCY BAND (Alpha, Beta). The statistical threshold was set up for a type 1 error rate of 5%. As effect sizes, we reported the proportion of explained variation (partial coefficients of determination, RP2). RP2 values were calculated using the ad hoc procedure for linear mixed-effects models implemented in the effectsize package [58]. Main and interaction effects were investigated post hoc using general linear hypotheses testing of planned contrasts from the multcomp package. We applied Holm’s sequential corrections to control the false discovery rate [59].

## 4. Results

Demographic information for each experimental group is provided in Table 1.

### 4.1. Change in Subjective Scores

#### 4.1.1. Anxiety Ratings

The TEST by GROUP interaction affected the subjective scores for perceived anxiety on the NRS (F(2,62) = 2.85, *p* < 0.05, Rp2 = 0.08, p1-β = 0.50) (Figure 2). Post hoc analyses revealed that the decrease in anxiety ratings from the pretest to the posttest in the foam rolling group (pretest: 3.15, 95% CI [2.53,3.77]; posttest: 1.95, 95% CI [1.33,2.57]) was greater than that recorded in the relaxation group (pretest: 2.00, 95% CI [1.45,2.55]; posttest: 1.48, 95% CI [0.93,2.03]) (*p* < 0.05). There was no difference between the foam rolling and manual massage groups (pretest: 2.60, 95% CI [1.98,3.22]; posttest: 1.45, 95% CI [0.83,2.07]), nor between the pretest to posttest differences observed in the manual massage, compared to the relaxation, group (*p* > 0.05).

#### 4.1.2. Muscle Relaxation, Warmth, and Pain Ratings

The TEST by GROUP interaction affected overall muscle relaxation ratings (F(2,62) = 4.67, *p* < 0.01, Rp2 = 0.13, p1-β = 0.48). Post hoc analyses revealed that participants reported a lower increase in perceived muscle relaxation from the pretest to the posttest in the foam rolling group (pretest: 5.30, 95% CI [4.65,5.95]; posttest: 7.20, 95% CI [6.55,7.85]), compared to the manual massage group (pretest: 5.25, 95% CI [4.60,5.90]; posttest: 8.75, 95% CI [8.10,9.40]) (*p* > 0.01). No such difference was present between manual massage and relaxation groups (pretest: 5.44, 95% CI [4.86,6.02]; posttest: 8.36, 95% CI [7.78,8.94]), nor between foam rolling and relaxation groups (both *p* > 0.05) (Figure 2).

The perceived muscle warmth was not affected by the GROUP by TEST interaction, but by a main effect of TEST (F(1,62) = 8.11, *p* < 0.01, Rp2 = 0.12, p1-β = 0.55). Posttest ratings (4.58, 95% CI [4.17,5.00]) were higher, compared to the pretest (5.26, 95% CI [4.85,5.68]) (*p* < 0.01).

The TEST × GROUP interaction affected perceived overall muscle pain scores on the NRS (F(2,62) = 11.00, *p* < 0.001, Rp2 = 0.26, p1-β = 0.62). Post hoc analyses revealed that participants experienced an increase in perceived muscle pain from the pretest to the posttest in the foam rolling group (pretest: 1.15, 95% CI [0.82,1.48]; posttest: 1.75, 95% CI [1.42,2.08]), whereas reduced pain scores were observed in the massage (pretest: 1.70, 95% CI [1.37,2.03]; posttest: 1.15, 95% CI [0.82,1.48]) and relaxation (pretest: 1.52, 95% CI [1.23,1.81]; posttest: 1.28, 95% CI [0.99,1.57]) groups (Figure 2).

### 4.2. Cognitive and Motor Performances

#### 4.2.1. Mental Calculation

There was no TEST by GROUP interaction and no main GROUP effect on the total number of correct responses. However, we found a TEST effect (F(1,62) = 24.82, *p* < 0.001, Rp2 = 0.29, p1-β = 0.80). The number of correct answers increased from the pretest to the posttest (pretest: 20.75, 95% CI [18.15, 23.36]; posttest: 23.57, 95% CI [20.97,26.17]) (*p* < 0.001).

#### 4.2.2. Sit-and-Reach Data

The GROUP by TEST interaction did not affect SRT performances but the linear mixed effects analysis revealed a main effect of TEST (F(1,62) = 44.85, *p* < 0.001, Rp2 = 0.42, p1-β = 0.95). SRT performance increased from the pretest to the posttest (pretest: 3.63, 95% CI [1.80,5.46]; posttest: 5.75, 95% CI [3.93,7.58]) (*p* < 0.001). There was no main GROUP effect (*p* > 0.05).

#### 4.2.3. Toe-Touch Data

TT performance was affected by the TEST by GROUP interaction (F(2,60) = 4.58, *p* < 0.01, Rp2 = 0.13, p1-β = 0.46). Post hoc analyses revealed that participants reported a greater increase in TT performance from the pretest to the posttest in the manual massage group (pretest: 2.11, 95% CI [−1.92,6.13]; posttest: 4.53, 95% CI [0.50,8.55]) than in the relaxation group (pretest: 3.13, 95% CI [−0.47,6.73]; posttest: 4.20, 95% CI [0.60,7.79]) (*p* < 0.01) (Figure 3). There was no difference between foam rolling (pretest: 7.74, 95% CI 3.72,11.76]; posttest: 9.53, 95% CI [5.50,13.55]) and manual massage groups, nor between foam rolling and relaxation groups (all *p* > 0.05) (Figure 3). The linear mixed-effect analysis also revealed that the main effect of TEST affected TT performance (F(1,60) = 86.70, *p* < 0.001, Rp2 = 0.59, p1-β = 0.99), by an increase from the pretest to the posttest (pretest: 4.23, 95% CI [2.00,6.46]; posttest: 5.94, 95% CI [3.70,8.17]) (*p* < 0.001).

### 4.3. Neurophysiological Data

#### 4.3.1. EEG Power Analysis

Spatial distributions of EEG power values in the sensors–space are provided in Figure 4. EEG power values were affected by the GROUP by TEST interaction (F(2,1412) = 3.33, *p* = 0.03, Rp2 = 0.02, p1-β = 0.99). Post hoc analyses revealed that the difference from the pretest to the posttest in the foam rolling group (pretest: 0.15 μV^2^, 95% CI [0.04, 0.25]; posttest: 0.16 μV^2^, 95% CI [0.06, 0.26]) was reduced compared to that observed in the relaxation group (pretest: 0.33 μV^2^, 95% CI [0.25, 0.41]; posttest: 0.29 μV^2^, 95% CI [0.20, 0.35]) (*p* < 0.03) (Figure 4). Likewise, the difference from the pretest to the posttest in the manual massage group (pretest: 0.23 μV^2^, 95% CI [0.18, 0.28]; posttest: 0.22 μV^2^, 95% CI [0.16, 0.27]) was reduced compared to the relaxation group (*p* < 0.05).

The linear mixed-effects analysis also revealed a main BAND (F(1,1412) = 453.89, *p* < 0.001, Rp2 = 0.18, p1-β > 0.95) and SENSOR effects (F(11,1412) = 6.13, *p* < 0.001, Rp2 = 0.24, p1-β = 1). Alpha power (0.33 μV^2^, 95% CI [0.29, 0.37]) was higher than Beta power (0.14 μV^2^, 95% CI [0.09, 0.18] (*p* < 0.001). Power analyses eventually revealed greater power values in frontal (FP1, FP2, F3, F4, and Fz sensors) than parietal regions (CP3, CP4, P3, and P4 sensors) (all *p* < 0.01).

#### 4.3.2. Skin Conductance

The linear mixed effects analysis revealed that the two-way interaction between TEST and GROUP affected the SCNorm (F(2,38) = 3.80, *p* = 0.03, Rp2 = 0.17, p1-β = 0.60). Post hoc analyses revealed that the pretest vs. posttest difference in the foam rolling group exhibited a decrease pattern (pretest: −0.39%, 95% CI [−12.83,12.04]; posttest: −12.86%, 95% CI [−25.30,−0.42]), whereas an increase pattern was present in the relaxation group (pretest: −3.20%, 95% CI [−14.89,8.49]; posttest: 16.20%, 95% CI [4.52,27.89]) (Figure 5). The difference between the foam rolling and the manual massage (pretest: 1.63%, 95% CI [−11.73,14.99]; posttest: 16.64%, 95% CI [1.20,26.09]) groups did not reach the statistical significance threshold (*p* = 0.12, NS), while there was no difference between manual massage and relaxation groups (*p* = 0.95, NS).

## 5. Discussion

This study compared the effects of FR and MM on subjective and objective indexes of performance and well-being. To the best of our knowledge, this is the first study comparing the effects of FR and MM in a single experimental design including EEG. Compared to the relaxation group, FR and MM groups exhibited distinct EEG patterns, as revealed by resting-state power values recorded in the sensors–space. All interventions were associated with improvements in range of motion and cognitive performances. FR and MM interventions had selective impacts on subjective ratings of well-being. MM positively impacted muscle sensations, whereas FR was associated with reduced general anxiety. It is noteworthy that the present trial was underpowered for effects with a low to medium effect size. A posteriori power calculation indicates that replication experiments should consider a sample of 30 participants per group to achieve adequate statistical power across all comparisons. Overall, present results add to the body of evidence suggesting that the acute effects of FR and MM benefit objective and subjective indexes of performance and well-being, despite their distinct demands in terms of motor control strategies [7].

Participants receiving MM remain passive and motionless, which promotes relaxation states but also emphasizes retroactive modes of action control [7]. On the contrary, FR requires voluntary actions and attentional focus on the components of the routine, which emphasizes proactive modes of action control [7]. Despite these fundamental distinctions, both interventions yielded increased synchronization of alpha and beta oscillations, compared to the relaxation condition. This resting-state pattern was early associated with a neurophysiological state of relaxation [26]. Our results first corroborate past reports indicative of relaxation brain states in response to massage experiments [28,60,61]. FR and MM interventions both involved mechanical pressures on soft tissues. Mechanical pressures on the skin, muscles, and fascia, exerted by the hands of a physiotherapist or an FR tool, activate interstitial receptors type III, IV, and Ruffini corpuscles. This can result in bottom-up homeostatic downregulations of the muscle tone [7,12,14,18]. The effect of mechanical pressures on sympathetic/parasympathetic regulation is a plausible account of EEG relaxation patterns [7,12,14,18]. The topography of alpha and beta power between the pre- and posttest further exhibited a distinct profile in the FR and MM groups. Synchronization was primarily detected from frontal sensors for FR and parietal sensors for MM. This is congruent with an assumption of homeostatic regulations in response to both types of intervention. Due to its proactive nature, FR might engage to a greater degree frontal region during the intervention, whereas MM might put a specific emphasis on parietal networks specialized in the sensory integration of peripheral feedbacks [62,63,64]. EEG data thus indicated that the proactive nature of FR could result in qualitatively different relaxation brain states, compared to MM. This is corroborated by skin conductance data, which revealed increased physiological arousal during the posttest compared to the pretest during FR, whereas reduced physiological occurred from the pretest to the posttest in the MM and autogenic relaxation groups. This again fits a homeostatic regulation pattern, where the respective proactive and retroactive demands of FR and MM are reflected in a respective upregulation of the sympathetic/parasympathetic branch of the autonomic nervous system [6,18]. The neural synchronization patterns observed as a result of FR and MM were absent in the autogenic relaxation intervention. Autogenic relaxation elicited a neural desynchronization pattern typically associated with perceptual and judgment tasks, rather than relaxation [49,65]. This could be due to the nature of cognitive resources mobilization and the participant’s inexperience of autogenic training, compared to MM and FR. Pfurtscheller and Lopes da Silva [49] emphasized that an increase of task complexity or attention results in an increased magnitude of neural desynchronization. Autogenic relaxation is a cognitively oriented (top-down) relaxation method, whereas MM and FR belong to autonomically oriented (bottom-up) relaxation techniques, in the framework for relaxation interventions laid down by Lehrer [66]. During this relaxation condition, participants pay specific attention to body sensations and breathing movements [67,68]. The autogenic training routine further involves various forms of motor imagery, i.e., the voluntary build-up of motor representations from proprioceptive and/or visual information stored within procedural memory [69,70]. Motor imagery involves cortical activation patterns during mental representation as well as during the subsequent resting-state periods [70,71,72]. Involvement of motor imagery during autogenic relaxation, but not during FR and MM, might thus account for the differences observed in the synchronization patterns of resting-state neural oscillations.

Motor performances indexed from range-of-motion measurements is a physical quality commonly targeted by massage interventions [73,74]. All groups improved their stretching performances. Possibly, the relaxation condition in the present study failed to depart from MM and FR, compared to a strict no-intervention condition. Indeed, while autogenic relaxation does not involve mechanical pressure on soft tissues, improvement in body awareness has the potential to reduce muscle tension and improve flexibility [70]. Autonomic regulation of the muscle tone indeed plays a key role in flexibility performance and was associated with neural synchronization patterns in past massage interventions [12,75]. Crommert et al. [76] observed, using ultrasound shear wave elastography, that the effects of massage interventions on muscle stiffness returned to basal level after 3 min. Here, the end of the massage intervention was separated from range of motion tests by 4 min, which were allocated to the resting state recordings. This time delay could have attenuated the effects of the intervention. As for SRT, TT performances increased from the pretest to the posttest in all groups. Yet, improvements in the MM group outperformed those recorded in the relaxation group, while there was no difference between the FR and relaxation groups. We thus postulate that the physiotherapist’s expertise represents a crucial factor for efficient stimulation of proprioceptive organs, compared to FR pressure [7]. Moraska [77] provided evidence that therapists with 950 h of didactic training achieved significantly better results than with 450 or 700 h of training. MM by a physiotherapist enables a greater variety of movements, thus promoting more precise regulation of pressure points in the three dimensions, compared to FR. The FR tools are limited to on-dimension trajectories and thus have a possibly reduced impact on autonomic regulations of muscle tone. We hypothesize that increased physical and cognitive demands associated with active FR, compared to passive MM and relaxation interventions, could have interfered with cognitive performances, as underlined in previous research (for an overview, see [7]). Performances on the mental calculation task improved across all experimental groups, which refutes our working hypothesis. Progress in the relaxation group prevents from drawing firm conclusions regarding the benefits of FR and MM on cognitive performances. Yet, improvements from the pre- to the posttest could purely reflect task habituation. Cognitively oriented relaxation routines, such as autogenic training, were indeed shown to improve cognitive performances [78]. MM and FR, which belong to the category of autonomically oriented relaxation methods, could hence be considered as equally beneficial as cognitively oriented relaxation methods on cognitive performances.

FR and MM both affected participants’ subjective reports. FR, but not MM, was further associated with reduced perceived anxiety. However, FR was also associated with increased pain and reduced perceived relaxation, compared to MM. MM thus appeared more effective than FR to improve overall perceived muscle sensations. This is in keeping with differences between FR and MM observed for range of motion and suggests a more efficient stimulation of the soft tissues during MM. The gestures of the physiotherapist appear to outperform self-administration of pressures through FR, although FR presents the advantage of greater reproducibility and reduced reliance on external intervention. Understanding why FR promoted a reduction of the perceived anxiety while concomitantly being associated with reduced beneficial effects on muscle sensations is not straightforward. A first hypothesis is that the proactive nature of FR emphasized the mind–body connection, which is a critical factor contributing to emotional release in massage therapy [79]. A more speculative hypothesis is that FR yields pressures up to 516 mm/Hg, which is fivefold greater than pressures elicited during MM (100 mm/Hg) [7,80,81] and was here associated with higher pain discomfort. Past research emphasized that pain perception is likely to shortcut the allocation of attentional resources to internal states [82]. Psychobiological models of fatigue emphasized that sensorial systems tend to allocate more attentional resources to the perception of effort than to external/environmental factors [83]. FR possibly acted as a distracting agent from the general anxiety state due to increase pain in response to more intense pressure stimulation.

As with all research, there are several limitations to consider before drawing final conclusions. First, the present design did not involve a no-intervention group. We prioritized controlling for the relaxation effects of the massage interventions by administering a cognitively orientated relaxation routine that did not involve pressure stimulation on the soft tissues [84,85,86]. Rather than a no-intervention group, we opted for a more discriminant form of control intervention, which appeared a higher methodological standard to discriminate the effects of autonomically oriented vs. cognitively oriented relaxation interventions [66]. Nonetheless, a no-intervention control group would enable to control for learning/task habituation effects from the pretest to the posttest. This particularly applies to the evaluations of cognitive and motor performances using a mental calculation paradigm and standardized flexibility tests where progress from the pretest to the posttests was present in all groups. This might reflect task habituation rather than the effect of the experimental interventions. However, habituation effects cannot account for interaction effects indicating distinct response profiles from the pretest to the posttest between experimental groups. A second limitation relates to the randomization procedure. The simple randomization yielded a different number of participants across groups. Simple randomization is preferable for experimental designs involving larger sample sizes, where it can be trusted to generate a similar number of participants per group [56]. Block randomization would have been a more appropriate methodological approach to allocate participants randomly to FR, MM, and relaxation groups while simultaneously minimizing sample size differences across groups.

## 6. Conclusions

FR and MM both positively impacted the flexibility of posterior chain muscles and cognitive performances on a mental calculation task. However, their effects remain difficult to dissociate from those resulting from a cognitively oriented relaxation routine that did not involve pressures on soft tissues. Although FR and MM groups exhibited distinct patterns profiles of resting-state activity, both interventions promoted brain states of relaxation, characterized by alpha and beta synchronizations in the frontal and parietal region, respectively [19,28,87,88,89]. Conversely, a desynchronization was observed after autogenic training. Similar to others before us, we suggest that the effects of MM and FR may involve autonomic nervous system regulations as a result of the bottom-up stimulation of proprioceptive organs as a result of mechanical pressures [12,13]. From a practical standpoint, present results seem to support the hypothesis that MM and FR are relevant to promote recovery in athletic populations [6,90,91]. FR and MM should, however, certainly not be restricted to high-level athletes. Sedentary people and patients suffering from depression/anxiety could benefit as well from regular MM and FR, considering their potential effects on subjective and objective indexes of performance and well-being [7]. Spurred by present findings, examining the therapeutic relevance of FR interventions in clinical populations definitely represents a promising research avenue.

## Figures and Tables

**Figure 1 brainsci-11-00818-f001:**
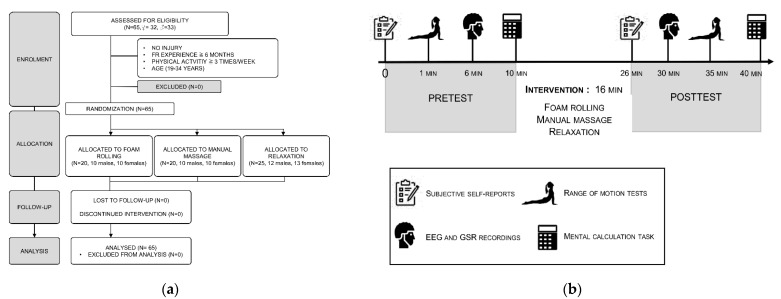
(**a**) Flowchart (inspired from CONSORT guidelines [32] describing participants’ inclusion and randomization procedures; (**b**) flowchart of the experimental procedures and measurements. GSR: galvanic skin resistance; EEG: electroencephalography.

**Figure 2 brainsci-11-00818-f002:**
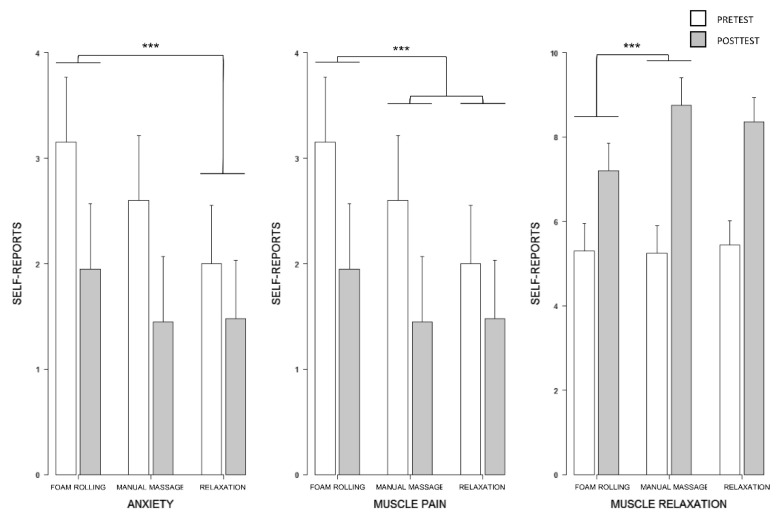
Barplot of the TEST × GROUP interaction effects obtained for subjective scores. Average fitted values by the linear mixed-effects analysis are presented with 95% confidence intervals (dotted bars). *** *p* < 0.001.

**Figure 3 brainsci-11-00818-f003:**
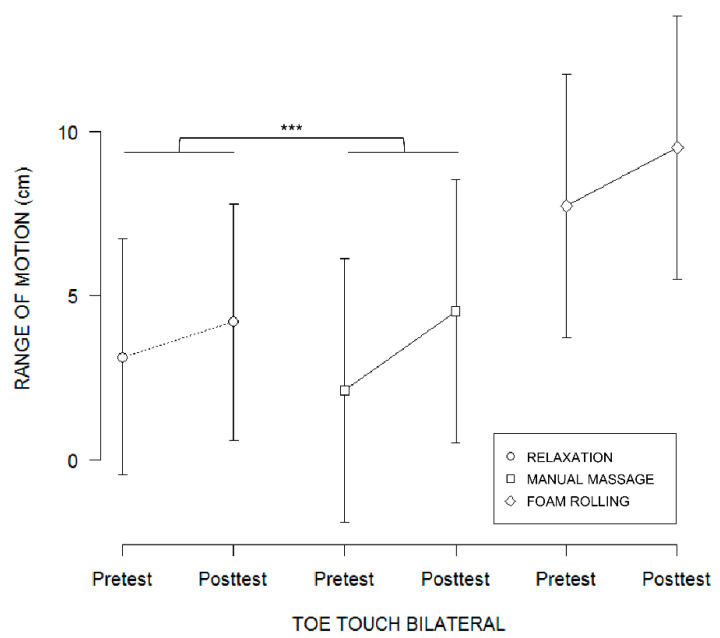
Fitted values for the TEST*GROUP interaction effect on range of motion revealed by the linear mixed-effects analysis, represented with 95% confidence intervals (error bars). *** *p* < 0.001.

**Figure 4 brainsci-11-00818-f004:**
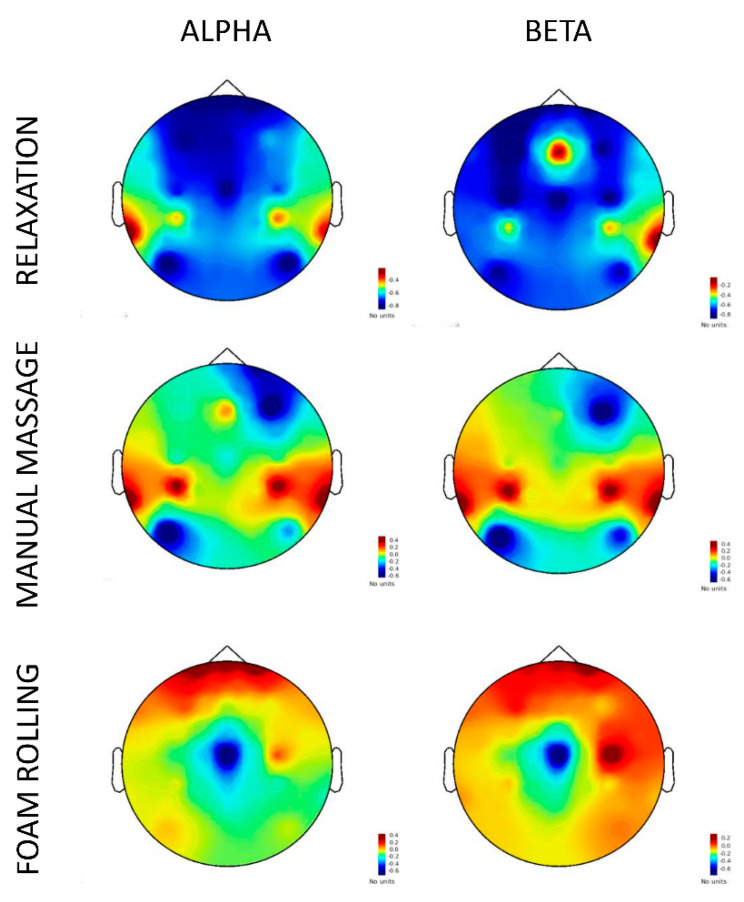
Spatial distribution within the sensors–space of the pretest vs. posttest normalized difference in EEGrs power across alpha (8–12 Hz) and beta (13–30 Hz) frequency ranges ((A − B)/(A + B)). Noteworthy, neural synchronization was primarily detected from parietal sensors for MM and frontal sensors for FR.

**Figure 5 brainsci-11-00818-f005:**
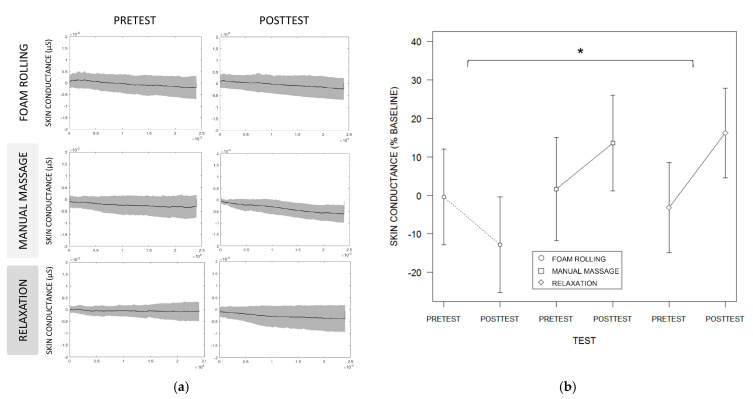
(**a**) Group average (SD) for the time course of skin conductance values during the pretest and posttest resting-state recordings; (**b**) fitted values revealed by the linear mixed effects analysis for the TEST × GROUP interaction on skin conductance, represented with their 95% confidence intervals (error bars). * *p* < 0.05.

**Table 1 brainsci-11-00818-t001:** Baseline characteristics (M ± SD). There were no adverse events and no subjects withdrew from the study. BMI: body mass index.

Characteristics	Age (years)	Height (m)	Mass (kg)	BMI (kg/m^2^)
**Relaxation**	25.9 ± 3.2	1.71 ± 10.5	67.3 ± 11.1	23 ± 2.3
	(range 20–34)	(range 1.53–1.91)	(range 48–98)	(range 19.8–28.8)
**Foam Rolling**	26.1 ± 3.1	1.73 ± 7	70.9 ± 10.4	23.6 ± 2.4
	(range 19–31)	(range 1.62–1.83)	(range 53–90)	(range 19.5–30.1)
**Manual Massage**	24.8 ± 3	1.72 ± 12.3	70.3 ± 13.4	23.6 ± 2.6
	(range 20–32)	(range 1.53–1.97)	(range 51–95)	(range 19–27.8)

## Data Availability

The datasets used and/or analyzed during the current study are available from the corresponding author on reasonable request.

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
