# Peer review of "Foam Rolling Elicits Neuronal Relaxation Patterns Distinct from Manual Massage: A Randomized Controlled Trial"

_brainsci, 2021, doi:10.3390/brainsci11060818_

Round 1
Reviewer 1 Report
I have several concerns about the presentation of the method, results and findings. In particular the authors are inconsistent about how they talk about the control group. Is the control group the participants that were assigned to the relaxation group--I thought so before I read the discussion section. The conclusions about the relaxation group are vast in the discussion and thus this does not seem like a control group. In addition, while the authors indicate that they FR and MM are similar, the results seem to bear out that they are not similar at all. Notably in the results section, they talk about analyzing Mu but in all other places they only discuss alpha and beta measures. In addition, it is unclear why beta is defined as 15-30 Hz rather than the standard 13-30 Hz. I would suggest the authors clarify this in the content of the manuscript as well as in the abstract and the discussion section. Again the conclusions very the most unclear section and needed information on limitations to the study also. In addition, page 10 mislabels frontal versus parietal areas--AND synchronization is not well defined in the manuscript.
Author Response
Please the response file enclosed.

Reviewer 2 Report
It is an interesting paper that assesses the topic in a rather comprehensive manner - so there is a lot to like here. Overall I like the design and the openness of the reporting, so I would be happy to recommend it for publication. I just have several suggestions I believe would make the paper better or more accurate.
- You use the terms "psychometric" and "behavioral" in an unexpected sense (at least to me). What you refer to as "psychometric" are actually self-report questionnaires assessing the emotional states. As all measures that assess any psy characteristic can be labelled as psychometrics (that is including mental calculation), I would strongly advise that you simply state what was measured as the outcome and that is: changes in affect (anxiety and relaxation), motor abilities(sit-reach and toe-touch), cognitive performance (mental calculation), physical pain (muscle) and resting-state EEG.
- I am not sure I fully understand the group allocation method - was it random group allocation? if yes how was the randomization perfomed? Where groups matched in some way or was it full-blind allocation? Why did you allocate 25 participants to the relaxation group and 20 to the other two groups? What is the rationale behind this?
- The numeric rating scale is a typical Likert-type scale, correct?
- Overall, I believe more emphasis should be put on the possibility that pre/post-test effects are due to the learning/training. This should be better incorporated in the interpretation of the results.
- The main effect here is basically a post-hoc comparison - I believe that should be highlighted so that the results are easier to follow.
These are all my comments, I hope that the author will be able to incorporate them in the manuscript. I am not submitting any additional comments to the edithor.
Author Response
Please find the response file enclosed.
